# Effect of Glass Filler Geometry on the Mechanical and Optical Properties of Highly Transparent Polymer Composite

**DOI:** 10.3390/polym14235179

**Published:** 2022-11-28

**Authors:** Ilya Kobykhno, Andrey Kiryanov, Victor Klinkov, Alla Chebotareva, Stanislav Evlashin, Dandan Ju, Yiyong Wu, Alexander Semencha, Huiyang Zhao, Oleg Tolochko

**Affiliations:** 1Peter the Great St. Petersburg Polytechnic University, Polytechnicheskaya, 29, 195251 St. Petersburg, Russia; 2Skobeltsyn Institute of Nuclear Physics, Moscow State University, Leninskie Gory, GSP-1, 119991 Moscow, Russia; 3Center for Materials Technologies, Skolkovo Institute of Science and Technology, Bolshoy Boulevard 30, Bld. 1, 121205 Moscow, Russia; 4Research Center of Space Physics and Science, Harbin Institute of Technology, Xida St. 92, Harbin 150001, China; 5School of Material Science and Engineering, Harbin Institute of Technology, Xida St. 92, Harbin 150001, China

**Keywords:** thermoplastic polyurethane, optical transparence composite, glass flake, glass fiber, mechanical properties, optical properties

## Abstract

In this work, we studied the influence of the geometry and degree of filling of glass dispersed particles on the optical and mechanical properties of flexible high-transmission composites, based on thermoplastic polyurethane. Glass spheres, glass flake and milling glass fiber were used as fillers. Studies of mechanical properties have shown that the introduction of any filler leads to a decrease in tensile strength and an increase in the elastic modulus of the composite material, however, with the introduction of glass flakes and milling glass fiber, a significant increase in the yield strength of the material is observed. The optical properties of composites with glass spheres decrease exponentially with an increase in the volume fraction of the filler. With an increase in the concentration of glass flakes and milling glass fiber to 10 vol.%, a sharp decrease in transmission is observed. With a further increase in concentration, the orientation of the filler along the film occurs, due to which the transmission in the visible range increases to values close to those of a pure polymer.

## 1. Introduction

The use of solar energy is becoming more and more common in our lives. The modern development of solar energy requires new materials creation for both photovoltaic cells and protective cover glasses [1,2,3]. The properties of cover glasses have a great influence on the performance of photovoltaic cells, primarily protecting them from mechanical damage and environmental influences [4]. A significant contribution to the efficiency of solar cells is made by optical losses associated with the absorption coefficient of protective glass [5].

Traditionally, hard glass with a thickness of about 100 μm is used as protective coating for solar cell. However, the use of such glass significantly increases the weight of the solar cell, since the weight of the cover glass can be up to 70% of the weight of the entire cell [6]. Since solar cells are the main source of energy in space satellites, reducing their mass is extremely important. Within the framework of the ROSA (Roll-Out Solar Array) project, a flexible roll-up solar panel was tested on the International Space Station and it was shown that for a cell with a power of 15 kW, a weight reduction is up to 33% and usable space can be saved up to is 75%, in comparison with folded rigid panels of the same capacity [7]. To date, there are many approaches to the creation of flexible cover glasses based on polyamides [8,9,10], various polysiloxanes [11,12,13] and composites based on them [14,15,16].

When creating optically transparent composites, it is extremely important to match the refractive indices of the polymer matrix and the selected type of filler, as this directly affects the transmission and haze based on Snell’s law [17]. There are various approaches to solving this problem, including the modification or functionalization of the chemical structure of polymers [18,19,20,21] and the creation of polymer nanocomposites [14].

To create optically transparent polymer composites, glass materials are the most common. In particular, composites based on glass fibers [19,22,23,24,25] and spheres [17,26] have been widely studied. Additionally, fillers in the form of glass flakes are of great interest; Dunlap in his work showed both theoretically and experimentally that their use significantly reduces scattering and optical distortion, and the main loss of transmission intensity is associated with effects arising on their edges [27]. Scarfe et al. studied the effect of glass flakes on the mechanical and optical properties of Polyvinyl Butyral and showed that, in addition to maintaining high transparency, the introduction of flakes can significantly improve the mechanical properties of the resulting composites [28]. In addition to glass fillers, a number of studies have investigated the possibility of using polymer ribbons and fibers as fillers, since this greatly simplifies the problem of matching the refractive indices of the matrix and filler [29,30].

Despite such a broad study of optical composites, today there are practically no data on the effect of the geometric parameters of the filler (particles, fibers and flakes) simultaneously on the optical and mechanical properties of the material, which would be obtained using a single polymer and would make it possible to evaluate the contribution of geometry. Therefore, in the presented work, we tried to study the influence of the geometry of the filler, namely glass spheres, glass fibers and flat glass flakes, and its volume fraction on the optical and mechanical properties of flexible thermoplastic polyurethane (TPU) based composite to create new protective glasses for flexible roll-up solar panels.

## 2. Materials and Methods

### 2.1. Materials

Thermoplastic polyurethane (TPU) Vitur TM1413-85, T_g_ = −50 °C, ρ = 1.177 g/cm^3^, Shore A 85, refractive index 1.528 was used as a polymer matrix for creating composites. As a filler, glass dispersed particles with different dimensions were used: 0D—glass sphere (GS), 1D—milling glass fiber (MGF), 2D—glass flake (GF). Glass spheres have average diameter of d = 90 μm, density—ρ = 2.6 g/cm^3^, refractive index—1.54. Glass flakes from ECR-glass (Glassflake Ltd., Leeds, UK) of 2 sites GF300M (thickness 3 μm) and GF100M (thickness 1 μm) with mean size D_50_ ≈ 105–130 μm, ρ = 2.6 g/cm^3^, refractive index 1.52. Milling glass fiber from E-glass with d ≈ 15 μm and length 200 μm, ρ = 2.6 g/cm^3^, refractive index 1.54.

### 2.2. Composite Preparation

Specimens of optical composite materials were fabricated in two stages. In the first step, 10–15 wt.% of TPU was dissolved in N,N-dimethylformamide and required quintaty of glass fillers were added in solution. In order to remove moisture and the sizing agent from the surface of the glass filler, it was preliminarily subjected to heat treatment at 300 °C for 2 h. Then, thin films-precursors with a thickness of ≈40 μm were fabricated by mechanical rolling of the solution on a PTFE substrate. In this case, the filler (MGF and GF) had oriented along the surface of the sample as shown in the Figure 1. At the second stage, a stack of thin films was placed in a mold (120 × 120 mm) and subjected to hot pressing at temperature of 180 °C for 10 min and pressure of 0.9 MPa. Thus, the samples with the thickness of 300–500 μm and fillers content of 5, 10, 16 and 31 vol.% (10; 20; 30 and 50 wt.% respectively) were obtained.

### 2.3. Test Methods

Tensile testing was carried out using a Shimadzu AGS-100 kNX universal testing machine with the strain rate of 0.01 s^−1^. To prevent damage to the sample in the clamps, overlays of pure TPU with a thickness of 500 μm were glued to it.

The microstructure was studied using a Phenom ProX scanning electron microscope.

Diffuse transmission spectra were measured on an SF-56 spectrophotometer (OOO Lomo-Spektr) with a PDO-6 integrating sphere in the scanning mode in the range of 400–1100 nm. The diffuse transmittance of the sample was measured first, together with the correction due to instrument scattering. The correction includes the amount of light scattered by the instrument that enters the integrating sphere during measurement of diffuse transmittance and is related to the percentage correction that takes into account the amount of light scattered by the instrument that hits the integrating sphere in the absence of a sample. The second step is diffuse transmittance measurement.

## 3. Results and Discussion

### 3.1. Mechanical Properties

Figure 2 shows experimental curves obtained from the tensile testing of TPU specimens filled with GS. It is clearly seen that the introduction and subsequent increase in the content of spheres in the composite material leads to a monotonous decrease in both the tensile strength and the yield strength. In this case, the elongation to failure varies within the limits of the experimental error. Separately, it should be noted that for pure TPU, after a deformation of 300%, a sharp increase in stress is observed, which indicates the orientation of polymer macromolecules along the loading axis. However, this effect is not observed on composite material samples, which may be due to the fact that the filler leads to the formation of defects in the structure of the samples during tension as shown in the Figure 3b,c and after the destruction of the polymer-glass adhesion boundary, the spheres do not participate in the deformation process, but only lead to a decrease in the effective cross section of the sample.

SEM images of samples of TPU-GS composites after their tensile testing are shown in Figure 2b,c. It can be seen that the spheres begin to be squeezed out of the sample, leading to the formation of defects. Most likely, this is due to a change in the thickness of the sample during its deformation. In this case, it is obvious that the resulting defects are the main reason for the decrease in the tensile strength of the material. With an increase in the filler content, an increase in both the number and size of defects is observed, which leads to a monotonic decrease in the ultimate strength of the composites.

The introduction of ground glass fibers into the composite leads to an increase in the yield strength and elasticity modulus of the material (Figure 3). At the concentrations of more than 16 vol.% stress-strain curve shows strain softening behavior after the specimen yields at ~9.7 MPa.

Although the decrease in yield stress is observed in case of GS-reinforced PU, the MGF composites show nearly 125% increase in the yield strength at its volume content of 32%. Due to a relatively high surface area to volume ratio, the milled fibers are closely spaced and provide significant resistance to the microplastic flow in the matrix, with increasing the yield strength of the composite at high volume fractions. This also indicates a relatively good adhesive bond between the fibers and the matrix.

Micrographs show that under uniaxial loading, the fibers are oriented along the loading axis and the characteristic “tracks” are formed. The size of the observed defects does not exceed lengths of the milled fiber in the defect probably because, after sample rupture, polymer relaxation and large reversible deformations occur, which, among other things, reduce the observed defect size. This leads to a decrease in the overall strength of the sample at the insignificant decrease of full deformation.

Figure 4 shows the tensile test curves for samples filled with glass flake’s with a thickness of 3 and 1 μm having SAV 750 and 1850 mm^−1^, respectively. Samples with both types of flicks show similar dependences, starting from 5 vol.% of flicks, the elastic modulus and yield strength increase sharply. With a further increase in concentration, their values also continue to increase. In this case, starting from 10 vol.% of flakes, the elongation to failure sharply decreases to values of ≈200%.

For samples filled with GF100M, a further increase in the filler concentration leads to the fact that the nature of the destruction of the samples becomes brittle, and the elongation to failure is ≈5%. In samples filled with GF300M, the transition to brittle fracture is observed only at the maximum degree of filling (31 vol.%), as well as significantly lower values of the yield strength. This difference may be due to increase of SAV ratio by more than two times. In that case the load transfer between the matrix and the filler is significantly improved, and also slows down the mobility of polymer chains, and polymer becomes incapable of transition to highly elastic deformation.

On the other hand, when stretched due to transverse strain (TPU has a Poisson’s ratio value greater than 0.4 [31]), a comprehensive compression of the filler occurs, which is accompanied by an increase in adhesion in the initial deformation area, an increase in friction between the polymer and flakes after the destruction of the adhesion boundary; but in this case, it can lead to damage to the polymer at the points of its contact with the faces of the filler. In this case, a decrease in the thickness of the flake leads to an increase in the number of faces, as well as the surface area, per unit volume of the filler.

Analysis of the surface of the samples after the test shows that when using GF300M (Figure 5) and concentrations up to 10 vol.%, there are “tracks” on the samples surface indicating the filler sliding inside of the matrix and associated with the deformation of the polymer around the filler particles after their detachment. However, with a further increase in concentration, they completely disappear. At the same time, such tracks are observed on TPU-GF300M samples up to a filler concentration of 16 vol.%.

A comparison of the results of the tensile testing of samples with different types of fillers (Figure 6) shows that glass spheres are the least effective in improving mechanical properties. This is because of a relatively high surface area to volume ratio (Table 1), GS relatively far away from each other when compared to the MGF or GF, and the interaction of stress fields around the particles remains minimal. Therefore, they have practically no effect on the elastic modulus of composites, only slightly increasing it, while leading to a monotonic decrease in the yield strength. At the same time, the elongation of the samples practically does not change.

The use of MGF has a much greater effect on the elastic modulus, increasing it by about an order of magnitude, from 20 to 260 MPa when filled at 32 vol.%. The yield strength of the material also increases significantly. However, the dependence of the yield strength on the volume fraction of MGF increases markedly up to 10 vol.%, after which it increases only slightly. A similar dependence of yield strength on concentration is demonstrated by samples filled with GF300M. The use of glass flakes with a thickness of 1 μm will make it possible to obtain an almost monotonous increase in the yield strength of the composite with the filler concentration, while it increases from 4.2 MPa, for unfilled TPU, to 14.3 MPa, for a sample containing 31 vol.% GF100M.

The dependences of the elastic moduli of samples filled with GF show an exponential growth with increasing filler concentration, while flakes with a thickness of 3 μm show slightly lower values compared to flakes with a thickness of 1 μm. This effect is probably related solely to the surface area to volume ratio.

### 3.2. Optical Properties

Particle-filled systems with significantly different mechanical properties have promising prospects for practical applications. When discussing optical composites constructed according to the noted principle, the key parameters that determine the optical characteristics of the composite will be the size of the filler, the difference in the refractive indices between the filler and the matrix, and the uniformity of the filler distribution over the volume. The noted parameters will determine the physical mechanism of the composite-light interaction. In general, the total optical loss can be the sum of scattering and absorption losses. Since optical glasses with a low absorption coefficient were used as fillers, the nature of the losses can be caused by structural imperfections (inclusions of submicron air bubbles or impurities).

Figure 7 shows the transmission spectra of a series of samples with different fillers and their volume fraction in the 400–1100 nm spectral range. The transmittance value is given for a thickness of 100 µm. The spectra show that the influence of the nature of the filler on the transmittance is different. Compared to the original polymer, the composite with glass spheres shows the greatest reduction in transmittance level. Figure 7 also shows the non-monotonic nature of the change in transmission spectrum with increasing filler volume fraction in the composite. In order to establish the nature of the observed experimental dependencies, the physical causes should be considered in more detail.

In a rough approximation when the refractive index of the matrix and the filler coincide completely at working wavelength, there is no scattering at the boundary between them. Therefore, to minimize scattering, the choice of the matrix-filler pair is carried out according to the correspondence of the refractive index values.

In the general case, the nature of scattering is determined by the interconnection between the size of the scattering particles and the radiation wavelength (*λ*). The effect of scatterers, which size is negligibly small with respect to the wavelength of the light, can be calculated using Rayleigh scattering, which is proportional to *λ*^−4^. The application of the Rayleigh scattering mechanism requires that the system meet a criterion based on a “size factor” that relates the relative refractive indices of the matrix and filler, and the size of the scatterer to the wavelength of interest.

The dimensionless size parameter of scatterer *x* (also called Rayleigh regime) may be parameterized by the following ratio:(1)$$x=\frac{2\pi rn}{\lambda }$$
where *r* is the scatterer radius, *λ* is the wavelength of light in vacuum and *n* is refractive index of matrix. The Rayleigh regime refers to scatterers with *x* < 1.

Table 2 shows the values of *x* parameter for the studied series and will be discussed later. It can be seen that none of the fillers satisfies the noted criterion. Based on the above, the Rayleigh scattering theory is not applicable for the objects under study due to the larger size of the scatterers than the wavelength of the incident radiation.

For scattering particles with geometrical size comparable with wavelength of radiation more complex Mie scattering theory must be used. Mie’s theory uses expansions in spherical harmonics to describe scattering, but which in fact is only valid for spherical particles [32]. For all other special cases that differ in shape from a sphere, as well as a rough surface and scatterer agglomeration, it is required to carry out a numerical solution or choose one of the described approximations based on the system parameters. At the limit of large particles with respect to wavelength, Mie’s theory becomes geometric optics (GO) or “large particle scattering”, where the amount of scattering is independent of wavelength.

The Rayleigh-Gans (RG) approximation is often used as a special case of the Mie theory to describe the scattering of dispersed particles in a matrix. It includes empirical modifications to describe interparticle interference and multiple scattering. The RG theory assumes that each volume element of a large particle scatters the incident light independently of the others, in accordance with the Rayleigh scattering theory. In other words, there is no multiple scattering inside the particle, and the incident light is not affected by the presence of other volume elements. The RG approximation is useful for modeling composites for optical applications that have stringent requirements for transmission loss due to scatter. This model is convenient and highly reliable, since for such applications only the peak transmission region of the material is of interest [27].

The key quantity parameter in Mie theory is the scattering efficiency *Q*, which is the ratio of the true scatter cross-section to geometrical cross-section. The scattering efficiency factors *Q* in optical materials depending on the nature of scatterer are presented at Table 2 [33,34].

The scattering efficiency factor *Q* is the ratio of the real scattering cross section to the geometric cross section and depends on scatterer size, relative refractive index and wavelength, which are included in the terms *m*, *x* and *p* [35].

The radiation scattered by the particle or inclusions is superposition of the radiation of individual scatterers taking the phase difference between them. The parameter which characterizes the scattering process is the phase shift of a diameteral ray *p*. The relationship between phase shift, refractive index ratio, and scatterer size is described by the following expression:(2)p=2x(m−1)

Strictly speaking, in our case, only GS is a spherical filler, since the simplified approach of the Mie theory is valid for spherical particles. However, if the values of the refractive indices of the matrix and the filler are close, as well as the absence of a significant intrinsic absorption of the filler (which is postulated by the manufacturer), the geometric pattern of radiation propagation will not change drastically. Therefore, this approach is applicable for a rough estimation of the effect of the filler dimensions, where the smallest of the linear dimensions was used as the scatterer diameter (for glass flakes and MGF it is thickness). We note the regular practice of such simplifications of the scatterer geometry in different ceramic and composite materials [33,34,35].

The calculated values of the parameters *x, p* and *Q* are shown in Table 3. In all prepared series the refractive index of matrix *n* and filler *n_f_* is very close, and the ratio *m = n_f_/n ≈* 1. It is clear that the value of *Q* increases with particle radius and that the amount of scattering also increases.

The calculated values presented in Table 3 indicate the following:due to the large value of *x*, the transition of radiation through the filler can generally be described by geometric optics, the beam is very weakly refracted at the phase boundary point (due to the fact that *m* is close to 1) and that the reflected intensity (radiation) is also very small (also because *m* is close to 1);according to *p* values, for all series the phase lag on the central ray passing through the scatterer along a full diameter and not scattered ray passing through matrix increased many times with increasing scatterer diameter;the greatest value of scattering efficiency factor *Q* was observed for sample GS, which is associated with its maximum geometric size;for a glass flake filler, there was no phase lag between the radiation transmitted in the matrix and the filler;the series with glass flake filler can be considered by large-size limit Rayleigh-Gans (Jobst) approximation.

The UV-VIS-NIR spectra of the composites with different fillers were obtained in compliance with ASTM D1003 using an integrating sphere. The resulting dependences of the level of diffuse transmission at wavelength 900 nm are shown in Figure 8.

Consideration of the dependences of the diffuse transmission of samples according to the nature of its change allows us to distinguish two groups of samples. For the first group, GS filler, the transmission decreases with an increase in the volume fraction of the filler. The second one is a decrease in the transmission value at a low filler content, which then changes to a slight increase and a tendency to a constant value.

To begin with, the dependences with minimal filler additions should be considered. It has the same tendency for all samples—a decrease in the initial transmission value of the “unfilled” polymer. In general, based on the measurement technique used (positioning the sample in front of the inlet of the integrating sphere), the decrease in the value of the diffuse transmission can be caused either by absorption, scattering or reflection.

As noted earlier, the absorption coefficient of the fillers is close to zero, so another reason for the decrease in the diffuse transmission should be considered. The reflection coefficient can indeed reduce the transmission value, but this effect is solely due to the surface, that is, an increase in the surface reflection of the samples with a filler. Thus, the only realistic reason for the decrease in transmission at the first filler additions is the increase in elastic scattering. Scattered light, unlike absorbed light, is not irrevocably lost from the system—it simply changes direction and is lost from a beam propagating in a certain direction—but contributes in other directions. The scattering increase may not lead to a decrease in the diffuse transmission in the case of multiple scattering of radiation in the bulk of the sample. This is possible as a result of registration by the photodetector; the photons scattered to the opposite direction of the radiation flux (backscattered photons).

If we compare the transmission value of the matrix and the composites containing 5% of filler, the different decrease in transmission values was measured: from 7% (to GS) to 2.5% (to GF100M).

The transmission of composite material filled with spheres particles based on equivalent composite model (ECM) can be calculated using the following expression [36,37]:(3)$$T_{GS}=T_{\max}\exp(-Q\varphi )$$
where *γ* is extinction coefficient, *r* is radius and *φ* is fraction of scatterers. The fraction of scatterers taking into account the size factor and the probability of placing spheres on a plane in mutually non-overlapping planes, in accordance with this, the *φ* = *N*^1/3^/4*r* [32]. *N* is a volume fraction of filler.

Figure 8 shows the experimental and calculated dependences of diffuse transmittance for all series of fillers.

It can be seen from the figure that for the GS the experimental values are lower than the calculated ones. The value of this discrepancy slightly increases with increasing filler content. The dependences obtained indicate the presence of an additional mechanism for reducing the level of diffusion transmission. In our opinion, the most probable cause is optical absorption losses due to the multiple reflection of radiation at the filler-polymer interface, due to the irregularity of the surface of the glass spheres. This assumption comes from the imperfection of the surface of the spheres.

Another reason is that in the calculation of factor *Q*, the average (most probable) particle size was used, however, the sizes of some particles could be larger, which would increase the value of *Q*. It is clear that the greater scatterer density is the greater the contribution of multiple scattering to the diffusion transmission will be. Thus, for samples filled with GS, an increase in the volume fraction leads to a decrease in diffuse transmission with an increase in the volume fraction of the filler, which is caused by the large diameter of the microspheres and, as a consequence, by the large value of the scattering factor *Q*.

For samples of MGF, GF100M and GF300M the experimental dependences are more complicated. For the marked series of samples, there is a local minimum of diffusion transmission, after which its growth is observed. It should be noted that the minimum area, taking into account the error for them, is about 10% of the filler.

The samples of MGF can be interpreted as the finite cylinder problem, which is not exactly soluble. First of all, this is due to the need to find the amplitude scattering matrix, taking into account the decomposition of the incident and scattered fields into components parallel and perpendicular to the planes determined by the axis of the cylinder and the corresponding wave normals [32]. The analytical solution can be obtained based on the Rayleigh-Gans approximation. Theoretical work has shown that finite cylinders with a length to diameter ratio of only about 5 can be closely represented as infinite cylinders. Therefore, many problems associated with particles, such as textile fibers, asbestos fibers, and even smoke particles combined in a chain structure can be adequately considered within the framework of the theory of infinite cylinders [38].

A simplifying position for considering scattering from a filler in the form of a cylinder can be the consideration of a parallelepiped. In this case, such an approximation can be used due to the absence of MGF absorption. Thus, an estimate of the magnitude of the transmission from the volume content of MGF can be obtained by the following formula:(4)$$T_{MGF}=T_{\max}\exp\left(-\frac{2QN^{1/3}}{\pi h}\right)$$
where *h* is cylinder height.

It can be seen from the figure that with a uniform distribution of the filler in the form of a cylinder, a monotonic dependence should be observed. The experimental values of diffusion transmission are located lower than theoretical. From general considerations, the presence of a minimum on the curve can be explained by an increase in losses when the fibers are located in mutually intersecting planes, that is, with overlap and contact with each other. The latter, of course, should lead to the appearance of additional defects at the boundary.

There is no unambiguous physical explanation for the location of the minimum in the region of 10 volume percent. However, it can be assumed that at a lower filler concentration the probability of their agglomeration and mutual intersection is small, and at a higher one, the technology of layer-by-layer formation of the composite assumes their strictly parallel distribution relative to each other. Thus, the probable reason for the increase in losses is the misorientation in the arrangement of fibers, their intersection, and the resulting multiple scattering at the fiber-matrix interface. It should be noted that in the case of densely packed fibers (>15 vol.%), the theoretical curve is quite close to the experimental values. The latter indicates an increase in the uniformity of the distribution of the filler with an increase in its content in the composite.

The problem of GF radiation scattering is similar to the problem with a cylinder, in the approximation that its thickness becomes much smaller (but still larger than the radiation wavelength) and the width increases many times over. In this regard, the general formula describing diffuse transmission retains its form (The heights are 1 and 3 µm for GF100M and GF300M, respectively).

In our opinion, the presence of a dip in the region of 10 μm is also associated with defects at the boundary of the filler surfaces, on which the main losses occur, including backscattering and absorption at the imperfection of the filler boundaries.

## 4. Conclusions

Finally, we may conclude that a method has been developed for obtaining optical composite materials with high transmittance and improved mechanical properties based on thermoplastic polyurethane and glass fillers of different geometry.

The use of glass spheres leads to a monotonic decrease in the tensile strength and yield strength of the composite, while not having a significant effect on the elastic modulus. Ground glass fibers and glass flakes can significantly increase both the yield strength and the elastic modulus of the composite. The use of thin flakes increases the elastic modulus of used TPU from 30 to ≈1500 MPa.

With the increase in the filler concentration and surface-to-volume ratio, relaxation processes in the polymer structure depressed significantly, which prevents the transition to forced highly elastic deformation.

Summarizing the results of the joint analysis of the experimental diffuse scattering spectra and the calculated ones, it was found that:for fillers that do not meet the Rayleigh size criterion x, the scattering efficiency factor Q very roughly describes the amount of scattered radiation, the values closest to the experimental values were obtained for highly filled non-spherical additives;the closest to the model values was the behavior of a spherical filler, while there were no inflection regions on the dependence, and the dependence was a monotonically decreasing function;despite the formal correspondence of the parameters (*p*, *m* and *x*) of the Rayleigh-Gans, approximation can be very limitedly applied to glass flakes, since the main mechanism that increases optical losses is backscattering at the boundaries of the fillers, as well as the geometric arrangement of the filler through mutual overlap (for MGF;the most acceptable filler for practical use is MGF, GF100M and GF300M with high volume fraction;glass sphere (GS) with a diameter of 90 μm significantly reduces the level of diffusion transmission, so this filler is less suitable to application in optical composites taking in account the relative decreasing of its mechanical properties;it is also of practical interest to establish the influence of the glass flake thickness on the position of the minimum on the transmission curve, which is the subject of further research.

## Figures and Tables

**Figure 1 polymers-14-05179-f001:**
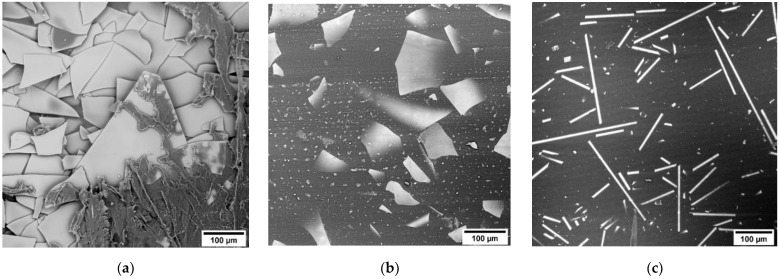
SEM images of the surface of precursor films (**a**); compacted samples with 31 vol.% of GF300 M (**b**) and 16 vol.% of MGF (**c**).

**Figure 2 polymers-14-05179-f002:**
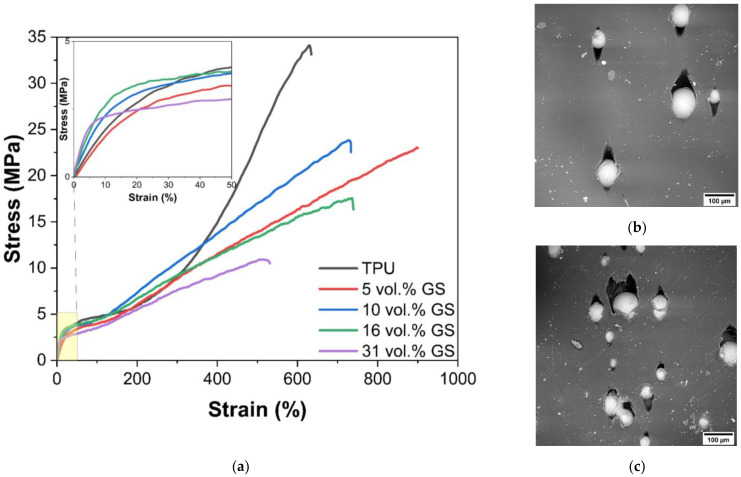
Stress-strain curves for samples with different GS content (**a**) and SEM images of the surface of samples with 10 (**b**), 31 vol.% (**c**) GS.

**Figure 3 polymers-14-05179-f003:**
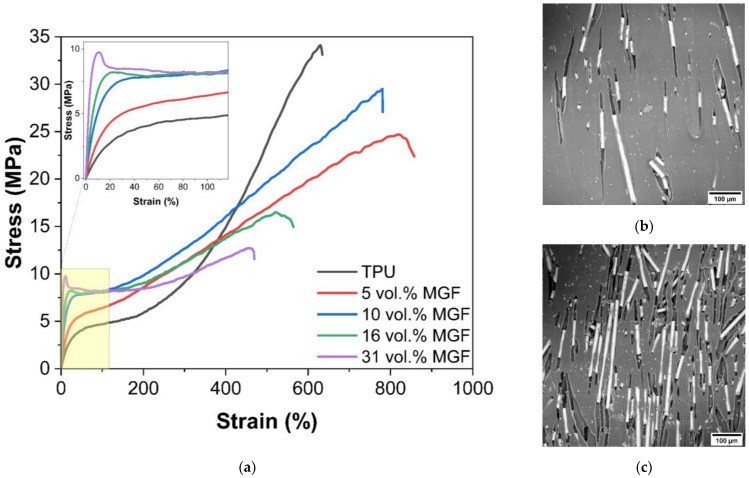
Stress-strain curves for samples with different MGF content (**a**) and SEM images of the surface of samples with 10 (**b**), 31 vol.% (**c**) MGF.

**Figure 4 polymers-14-05179-f004:**
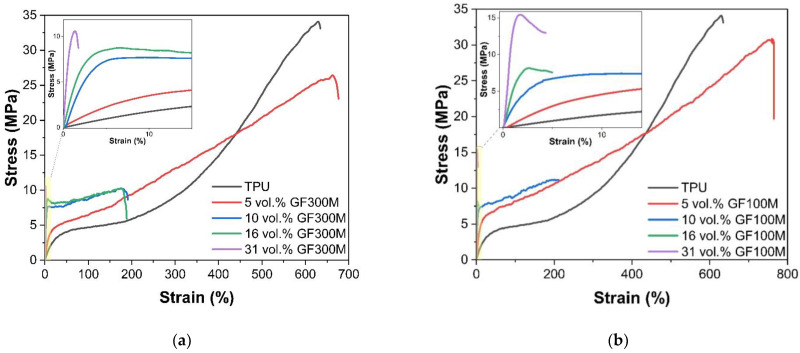
Stress-strain curves for samples with different content of GF with the thickness of 3 (**a**) and 1 (**b**) μm.

**Figure 5 polymers-14-05179-f005:**
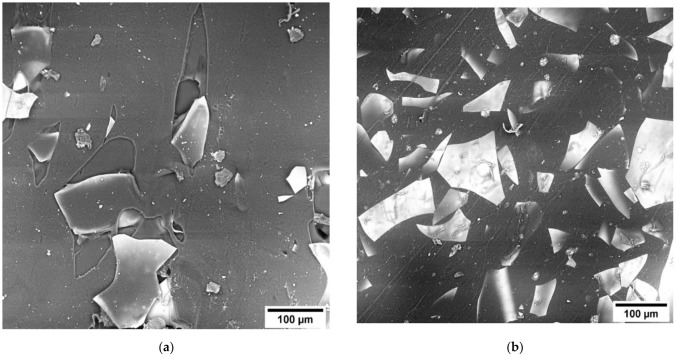
SEM images of the surface of samples with 10 (**a**) and 31 vol.% (**b**) GF300M.

**Figure 6 polymers-14-05179-f006:**
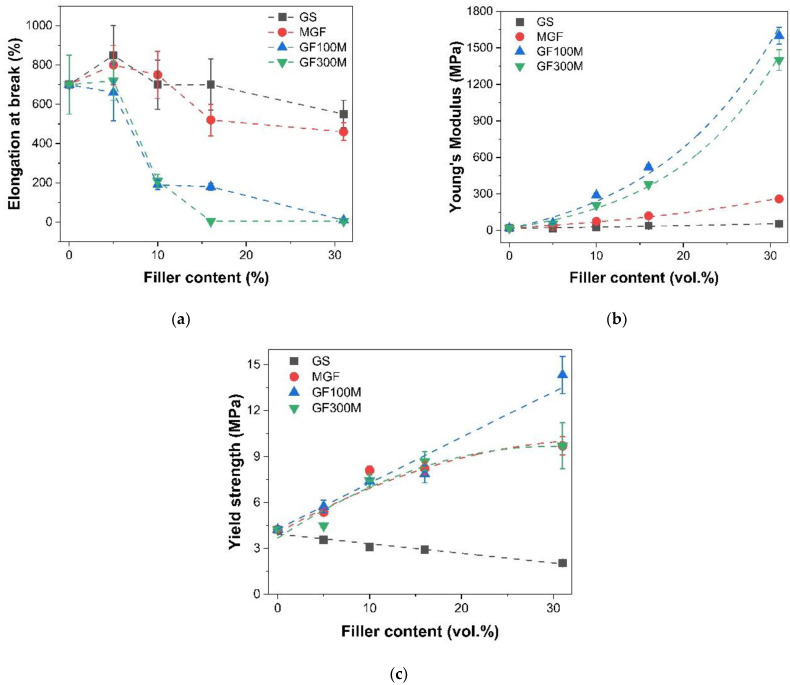
Dependency of elongation at break (**a**), Young’s modulus (**b**) and yield strength (**c**) from different types of the filler.

**Figure 7 polymers-14-05179-f007:**
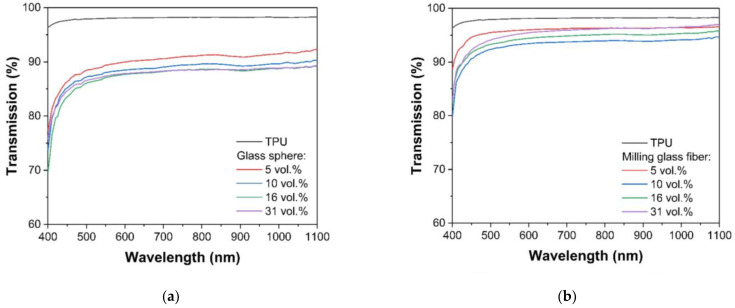
Transmission spectra of composite filling by glass sphere (**a**), milling glass fiber (**b**), glass flake GF100M (**c**) and GF300M (**d**).

**Figure 8 polymers-14-05179-f008:**
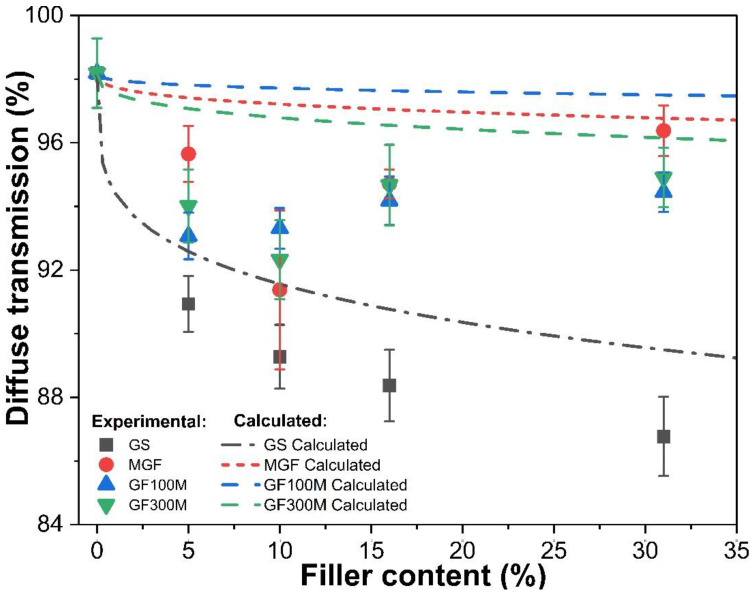
Concentration dependances of diffuse transmittance of TPU with different fillers.

**Table 1 polymers-14-05179-t001:** Surface area to volume ratio (SAV) for different filler type.

Filler Type	SAV, mm^−1^
GS	0.066
MGF	210
GF100M (*h* = 1 μm)	1850
GF300M (*h* = 3 μm)	750

**Table 2 polymers-14-05179-t002:** Scattering efficiency factors *Q*.

Scatterer		*Q*	Scattering Behavior
Intermediate size, *m* ≈ 1	*p* << 1	$p{\left(m-1\right)}^{2}x^{2}=\frac{p^{2}}{2}$	*λ*^−2^ Rayleigh-Gans
*x* > 1, *m* ≈ 1	1 < *p* << 1	$2-\frac{4}{p}\sin p+\frac{4}{p^{2}}(1-\cos p)$	
	*p* >> 1	*Q* > 2	Wavelength independent
	*p* < 1	$Q\to \frac{p^{2}}{2}=2{\left(m-1\right)}^{2}x^{2}$	*λ*^−2^ Jobst approximation (large-size limit of the Rayleigh-Gans approximation)

**Table 3 polymers-14-05179-t003:** Calculated values of *x*, *p* and *Q* for investigated samples.

Filler	*r*, µm	*x*	*p*	*Q*
GF100M	0.5	8.0	0.08	0.004
GF300M	1.5	24.0	0.25	0.031
MGF	7.5	120.1	1.88	1.455
GS	45	720.0	11.31	2.358

The calculation was made for *λ* = 0.6 µm.

## Data Availability

Not applicable.

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
