# Peer review of "Effect of Glass Filler Geometry on the Mechanical and Optical Properties of Highly Transparent Polymer Composite"

_polymers, 2022, doi:10.3390/polym14235179_

Round 1

Reviewer 1 Report

This paper describes the influence of the geometry and degree of filling of glass dispersed particles on the optical and mechanical properties of flexible high-transmission composites based on thermoplastic polyurethane.  The authors studied the influence of the geometry of the filler, namely glass spheres, glass fibers and flat glass flakes, and its volume fraction on the optical and mechanical properties and found that the most acceptable filler for practical use is MGF, GF100M and GF300M with high volume fraction.  I think the experiments were carefully done and the measured date are reliable.  The results of this paper will give useful information in the field of optically transparent polymer composites for photovoltaic cells.  I would like to accept this manuscript in Polymers.

May I have comments.

- The discussion of Figure 7 is missing.  The description at 3.2. Optical properties is that of 3.1 Mechanical properties.  Probably, the authors made a mistake in copy and paste process.

- I am just curious about the reason why the authors have interested in wavelength of 900 nm in Figure 8.

Author Response

1. The discussion of Figure 7 is missing. 

A description of Figure 7 has been added to the text of the article. We have made the appropriate changes.

2. The description at 3.2. Optical properties is that of 3.1 Mechanical properties.  Probably, the authors made a mistake in copy and paste process.

Thank you very much for the comment, there was indeed a technical error in transferring the text to the template. We have made the appropriate changes.

3. I am just curious about the reason why the authors have interested in wavelength of 900 nm in Figure 8.

The 900 nm wavelength was chosen for the following considerations: scattering is most pronounced when the magnitude of inhomogeneities, inclusions or impurities is comparable to the incident wavelength. With the introduction of a filler into the polymer the number of elementary scatterers at the polymer-filler interface inevitably increases and their geometrical dimensions are difficult to estimate experimentally. Figure 7 shows that in the spectral region up to 650 nm there is a significant change in transmittance and the slope of the curves increases, which also indicates scattering. For wavelengths longer than 900 nm the transmittance is less influenced by the nanoscale and therefore this area can be more correctly considered for the comparison of samples with different sizes.  

Reviewer 2 Report

The article entitled "Effect of glass filler geometry on the mechanical and optical properties of highly transparent polymer composite" is written very well and correctly. I have no reservations about the English language. The article can be read very smoothly and I do not find any errors in it. The authors raised a very interesting topic. I have just two attentions:

1:

Bad annotation in line 191:

“Table 1. This is a table. Tables should be placed in the main text near to the first time they are cited.”

2:

The graphs in Figure 6 should be enlarged and have too low resolution

The article may be published after minor corrections.

Author Response

1: Bad annotation in line 191: "Table 1. This is a table. Tables should be placed in the main text near to the first time they are cited.”

Thank you very much for the comment. We made the appropriate changes.

2: The graphs in Figure 6 should be enlarged and have too low resolution

Thank you very much for the comment. We made the appropriate changes.